# Harnessing Greenhouse Gases Absorption by Doped Fullerenes with Externally Oriented Electric Field

**DOI:** 10.3390/molecules27092968

**Published:** 2022-05-06

**Authors:** Rodrigo A. Lemos Silva, Daniel F. Scalabrini Machado, Núbia Maria Nunes Rodrigues, Heibbe C. B. de Oliveira, Luciano Ribeiro, Demétrio A. da Silva Filho

**Affiliations:** 1Institute of Physics, University of Brasília, Brasília 70919-970, Brazil; 2Laboratório de Modelagem de Sistemas Complexos (LMSC), Instituto de Química, Universidade de Brasília, Brasília 70919-970, Brazil; daniel.scalabrini@unb.br; 3Grupo de Química Teórica e Estrutural de Anápolis, Campus de Ciências Exatas de Anápolis, Universidade Estadual de Goiás, Anápolis 75132-903, Brazil; nubiamarianunes@gmail.com (N.M.N.R.); lribeiro@ueg.br (L.R.); 4Laboratório de Estrutura Eletrônica e Dinâmica Molecular (LEEDMOL), Instituto de Química, Universidade Federal de Goiás, Goiânia 74001-970, Brazil; heibbe@ufg.br

**Keywords:** C_20_ fullerene, C_19_Si, doping, external electric field, carbon monoxide, sensor

## Abstract

In this work, a theoretical investigation of the effects caused by the doping of C_20_ with silicon (Si) atom as well as the adsorption of CO, CO_2_ and N_2_ gases to C_20_ and C_19_Si fullerenes was carried out. In concordance with previous studies, it was found that the choice of the doping site can control the structural, electronic, and energetic characteristics of the C_19_Si system. The ability of C_20_ and C_19_Si to adsorb CO, CO_2_ and N_2_ gas molecules was evaluated. In order to modulate the process of adsorption of these chemical species to C_19_Si, an externally oriented electric field was included in the theoretical calculations. It was observed that C_19_Si is highly selective with respect to CO adsorption. Upon the increase of the electric field intensity the adsorption energy was magnified correspondingly and that the interaction between CO and C_19_Si changes in nature from a physical adsorption to a partial covalent character interaction.

## 1. Introduction

Carbon nanostructures have been continuously studied for the most diverse applications [1,2,3,4,5,6]. Among these nanostructures, fullerenes [7,8] stand out because of their good physicochemical reactivity and their symmetrical and relatively simple structure [9,10]. These characteristics render fullerenes, whether in their pristine or doped form, an attractive molecular system to be object of several theoretical studies aiming at various sorts of applications [11,12,13,14,15,16].

Among the fullerenes, C_20_ is the smallest known example containing only twelve pentagonal faces. Since the existence of this allotrope of carbon was already predicted [17], even before its experimental discovery [18], C_20_ and its derivatives already had an extensive history of studies [19,20,21,22].

Considered the most reactive of the fullerenes [18], interest in C_20_ has continued over time [23,24,25,26,27]. This molecule has been the object of several studies that aim to carry out its doping to investigate aspects related to its electronic and structural characteristics, in addition to proposing possible applications in different areas.

In a recent theoretical investigation, Metin and co-authors [15] used calculations with the theoretical chemistry model B3LYP/6-31G(d) to investigate both the hydrogen storage capacity and the electronic properties of C_20_, C_15_M_5_ and H_2_@C_15_M_5_, with M = Al, Si, Ga, Ge. Among the conclusions of the research, the authors highlighted that, fullerenes doped with Si and Ge, more specifically in the form C_15_Si_5_ and C_15_Ge_5_, were highly sensitive to the presence of H_2_. In addition, the ability of C_20_-doping to manipulate the physicochemical and structural parameters of fullerenes was also confirmed.

The analysis of C_20_, in its pure form and doped with aluminum (C_20−n_Al_n_; *n* = 1–5), was recently performed by Hassanpour and coauthors [23]. In a theoretical effort, employing various levels of Density Functional Theory (DFT) calculations, they investigated the Al substitution in the C_20_ cage. Through analysis of the infrared spectrum (Infrared-IR), the authors observed that both the number and the position of Al atoms can change the IR spectrum. Furthermore, they observed that by controlling the number of dopants, the concentration and distribution of atomic charges (APT charges) can be altered to favor adsorption of chemical species to the C_20−n_Al_n_ cage.

Amongst the various atoms possible for the doping of C_20_, Si has drawn attention in recent years. The Si atom belongs to the same family as carbon and may have interesting properties when used as an impurity in the cage of carbon nanostructures. Doping of C_20_ with Si was theoretically studied by Koohi, Amiri and SHariatI [28], using DFT calculations, the authors analyzed the effects of doping C_20−n_Si_n_ with *n* = 1–10. This strategy indicates that, in the same vein as in the case of Al doping, controlling the position and number of Si heteroatoms makes it possible to modulate structural changes of the molecule as well as the energies of the molecular orbitals. Similar results were also observed theoretically by Ajeel and co-authors [29].

Beside the aforementioned examples, there are interest in applying C_20_ and its derivatives for the most varied purposes, whether for hydrogen storage [15], as a drug delivery system [30,31] or for the detection of biomolecules [32]. A very promising application of these carbonaceous systems is in detection and capture of many gases [14,33] especially those associated with the greenhouse effect and/or harmful to health.

In this spirit, the present work is dedicated to the study of C_20_ and C_19_Si for the adsorption of carbon monoxide (CO) and carbon dioxide (CO_2_). Both CO and CO_2_ are products of burning fossil fuels. While CO_2_ is one of the main gases responsible for the greenhouse effect, along with methane gas, CO contamination is considered by the World Health Organization as one of the main causes of accidental poisoning around the world each year [34].

Theoretical DFT-based calculations were carried out to investigate the effects of the interaction between C_19_Si and CO and CO_2_ molecules. Recently, there has been much attention to the use of externally oriented electric fields (EOEF) to harness physical and chemical properties of molecular systems [35,36,37]. Intermolecular interactions between weakly interacting systems can be strongly aided by EOEFs, since it has the advantage of being easy to achieve and control, and is environmentally friendly [38]. On the realm of molecular sensors, previous works [16,39,40,41,42,43], have demonstrated that the adsorption of molecules and the storage of H_2_ can be facilitated in the presence of an EOEF. Additionally, EOEF might render selectivity to the sensor towards a given species in complex gaseous mixtures [16,39].

Based in our interest in theoretical investigations aiming at the development of new sensors [44], we disclose herein the influence of an externally oriented electric field to rationally control the adsorption process of gas molecules on doped C_19_Si fullerenes relying on DFT calculations. Since approximately 78% of the Earth’s atmosphere is made up of nitrogen gas (N_2_) [45], the selectivity of C_19_Si for the detection of CO and CO_2_ was evaluated by comparing the adsorption energy between the C_19_Si cage and the CO, CO_2_ and N_2_ molecules.

## 2. Materials and Methods

To confirm the structural and electronic trends observed in the fullerene cage after doping with a silicon atom, an investigation involving several theoretical levels was carried out. In all calculations, unrestricted optimizations were performed. The absence of negative frequencies confirmed that the molecules are at their energetic and structural minimum at each theoretical level considered. Several combinations of Exchange Correlation DFT functionals and basis sets (ωB97XD/6-31G(d), ωB97XD/6-311+G(d,p), ωB97XD/def2TZVP, M062X/6-31G(d), M062X/6-311+G(d,p), M062X/def2TZVP, M06L/6-31G(d), M06L/6-311G(d,p), M06L/def2TZVP, B3LYP/6-31G(d), B3LYP/6-311+G(d,p) and B3LYP/def2TZVP) were tested to screen electronic and structural properties of pristine and doped fullerenes. The screening of the theoretical model on the electronic and structural properties revealed the following combination of DFT Exchange-correlation function and basis set, ωB97XD/6-31G(d) and ωB97XD/6-311+G(d,p), as the optimal choice for the investigation of the dimers interaction considering the compromise of accuracy and computation cost. While the 6-31G(d) basis set has been reported to be a good choice for obtaining fairly reliable results in fullerenes [11,46], using the more extended 6-311+G(d,p) basis set improves the description of the non-covalent interactions, for further analysis of Quantum Theory of Atoms in Molecules (QTAIM) parameters and Reduced Density Gradient (RDG) properties as reported in recent studies [47,48].

Adsorption energies under electric field F, were calculated using the supramolecular approach considering the optimized structures by the expression [49],
(1)$$E_{ads}\left(F\right)=E_{gas-C_{19}X}\left(F\right)-\left[E_{gas}\left(F\right)+E_{C_{19}X}\left(F\right)\right];\;X=Si\;or\;C.$$

In Equation (1), $E_{ads}\left(F\right)$ is the adsorption energy. $E_{gas-C_{19}X}\left(F\right)$ refers to the energy of the dimer formatted by the C_19_X and the gas molecules, $E_{gas}\left(F\right)$ is the energy of the gas molecule and $E_{C_{19}X}\left(F\right)$ is the energy of the fullerene C_20_, for *X* = C or C_19_Si for *X* = Si.

The adsorption energies were corrected for the Basis Set Superposition Error (BSSE) using the Counterpoise method [50]. The calculations included EOEF, F, in atomic units (a.u.) (1 a.u. = 51 V/Å) in the range: *F* = 0.000 a.u., 0.001 a.u., 0.005 a.u., 0.010 a.u., 0.020 a.u. and 0.025 a.u. along the Si heteroatom (see Appendix A for clarity).

The description of intermolecular interactions was performed by means of the QTAIM [51,52,53] and RDG analysis [54,55]. Molecular Electrostatic Potential (MEP) map, electronic and structural properties were calculate with the Gaussian 16 [56] software. QTAIM and RDG properties were calculated using Multiwfn [57] wave function analysis program. VMD software version 1.9.3 [58] was employed to render isosurfaces and molecular representations.

## 3. Results and Discussion

### 3.1. Doping Effects on the Electronic and Structural Properties of Fullerenes

The structure of doped fullerene was constructed by replacing a carbon atom for a silicon atom. Initially, the molecules had their structures optimized at the ωB97XD/6-31G(d) level of theory followed by another round of calculation at the ωB97XD/6-311+G(d,p) model chemistry. Both pure and doped fullerene are shown in Figure 1 and the respective coordinates of these molecules are shown in the Appendix A.

From Figure 1 we clearly note that Si-doping promoted remarkable structural deformations of the fullerene cage. Due to its larger atomic radius, it was observed that the Si-C bond distances with the adjacent C-atoms was lengthened with respect to the C-C bond distances prior to the substitution. From a polarization perspective, Merz-Kollman (MK) charge analysis (using the 6-31G(d) basis set and in parenthesis with 6-311+G(d,p)) indicates that C_20_ has a modest charge distribution ranging from −0.018 (−0.103) to 0.018 (0.103)e (Figure 1D). For the C_19_Si, MK charges range from −0.341 (−0.366) to 0.341 (0.366)e (Figure 1E). Still on Figure 1D, it was also noted charge accumulation of −0.222 (−0.162)e on the Si-atom. MEPs shown in Figure 1D,E shows that, Si-doping imparts a polarization of the fullerene, with a dipole moment of C_19_Si of μ = 1.950 D (1.460 D) similar to what is observed in reported studies in literature [25,29]. As an immediate result of the polarization of the molecule, the effective intermolecular interaction can be strongly altered [11,59,60,61], so that C_19_Si display the ability to adsorb different molecular species aided by electrostatic forces (dipole-dipole, for instance) that was once absent in its pristine form.

Doping the C_20_ fullerene had direct impact on the electronic structure. Looking at the frontier eigenstates (HOMO and LUMO), shown in Appendix A, addition of an Si-atom induced an increase in the HOMO-LUMO energy gap, $E_{HL}$ ~0.3 eV: for C_20_ fullerene, $E_{HL}=5.46\;\mathrm{eV}\;$(6-31G(d)) and, $E_{HL}=5.40\;\mathrm{eV}\;$(6-311+G (d,p)) whereas for C_19_Si DFT calculations delivered $E_{HL}=5.81\;\mathrm{eV}\;$(6-31G(d)) and $E_{HL}=5.72\;\mathrm{eV}\;$(6-311+G(d,p)). Thus, it is noted that the increase in $E_{HL}$ is a trend observed for both bases used. It is important to highlight that, in previous works, the doping of nanocarbon structures, including fullerenes C_60_ [62,63] and fullerenes C_20_ [64,65], led to a decrease in the value of $E_{HL}$, which differs from the results observed in this work. Additionally, regarding the variation of the energy gap between the frontier orbitals, it is observed that the negative MK charges on the Si atom (see the MEP in Figure 1E), are not in line with the atomic charge on the heteroatom observed in the literature for C_19_Si [14,30].

To investigate whether such contradictions is an artifact of the level of theory employed, we performed additional unrestricted optimizations with different Exchange-Correlation DFT functionals (XCF) and basis set (see Materials and Methods section) keeping track to the absence of negative frequencies. The results of such extended theoretical survey on the frontier orbitals, as obtained at the B3LYP/6-31G(d), B3LYP/6-311+G(d,p), B3LYP/def2TZVP, ωB97XD/6-31G(d), ωB97XD/6-311+G(d,p), ωB97XD/def2TZVP, M06L/6-31G(d), M06L/6-311+G(d,p), M06L/def2TZVP, M062X/6-31G(d), M062X/6-311+G(d,p) and M062X/def2TZVP are reserved in Appendix A found in the SM file to avoid proliferation of tables in the main text. Glancing at Appendix A we report the percentual increase of HOMO-LUMO energy gap upon Si-doping, $\Delta E_{HL}$, and straightforwardly testify that regardless of the XCF/basis set combination, the addition of a Si-atom increases the chemical stability of C_19_Si over C_20_. Similarly, the apparent opposite findings of this work and those reported in literature [14,30] concerning the charge accumulation on Si-atom was retained for each of the screened XCF/basis set (see Appendix A). From Appendix A, partial MK charge on Si-atom, evidently is affected by the choice of XCF/basis set, nevertheless, all theoretical calculations consistently delivered a negative charge on Si. We, therefore, believe there is no dispute concerning this issue. Still on Appendix A we note that the different levels of calculations provoke slightly changes on bond lengths supporting our choice for ωB97XD/6-31G(d) and ωB97XD/6-311+G(d,p) as a reliable theoretical model for further analysis. Moreover, despite these contradictions with the results reported in some previous works, in which C_20_ was doped with Si [14,27,30], our calculations is in agreement with other similar works [24,25,29].

To further investigate the possible causes of the difference in results observed in these calculations with the results presented in the literature [14,27,30], and inspired by studies with the doping of C_20_ with Si and Al [23,28,29] we studied the impact of the position of Si-atoms in the fullerene cage. For this purpose, 20 configurations of C_19_Si were produced, named C_19_Si (*X*), where *X* = 1–20 indicate the label of the carbon atom replaced by a Si-atom, as depicted in Figure 2. All C_19_Si (*X*) were optimized without restrictions at the ωB97XD/6-311+G(d,p) level of theory.

Our results confirm that the doping of C_20_ with a Si atom generates a structural and electronic variation that depends on the position in which the impurity is inserted into the fullerene cage. However, only two types of C_19_Si geometrical conformations were retrieved. To simplify the discussion, these sets of substitutions will be referred to as the C_19_Si(A) set and the C_19_Si(B) set. Both sets are represented by Figure 2C,D, respectively.

The C_19_Si (A) set contains the following doped fullerenes: C_19_Si (1), C_19_Si (2), C_19_Si (4), C_19_Si (6), C_19_Si (7), C_19_Si (8), C_19_Si (11), C_19_Si (12), C_19_Si (13), C_19_Si (14), C_19_Si (15), C_19_Si (16), C_19_Si (17), C_19_Si (18) and C_19_Si (19). The C_19_Si (B) set is formed by the doped fullerenes C_19_Si (3), C_19_Si (5), C_19_Si (9), C_19_Si (10), C_19_Si (17) and C_19_Si (20). Looking at Figure 2 and Table 1, the C_19_Si (A) and C_19_Si (B) sets have similar members with close structural similarity, and the same tendency of charge accumulation on the Si atom, the same values of $E_{HL}$ in addition to the same total energy.

Thus, it can be seen in Table 1, that the $E_{HL}$ energy also presents different values depending on the position of the heteroatom. If the heteroatom is positioned to generate geometrical conformations of the C_19_Si (B) set (Figure 2D), $E_{HL}$ tends to increase. On the other hand, if the heteroatom replaces one of the carbons as indicated in C_19_Si (A) (Figure 2C), $E_{HL}$ tends to decrease. Considering that the variation of $E_{HL}$ is an indication of chemical stability in which higher values of $E_{HL}$ indicate a reduction in the chemical reactivity of a given molecule [66], we note from Table 1, that the geometrical conformers pertaining to C_19_Si (B) set are more chemically stable than those of the C_19_Si (A) set. Relying on B3LYP/3-21G calculations, Ajeel and coworker [29] did not observe significant changes in the $E_{HL}$ energy and in the structural properties of C_19_Si. Since only three out of ten C_19_Si geometrical conformers belong to the C_19_Si (B) set, a non-rational choice of doping site increases the odds of producing a C_19_Si (A) geometrical conformer.

Again, in Table 1, total electronic energy $E_{t}$ of the molecules was also considered to understand their relative stability. For the geometrical conformers of the C_19_Si (A) set $E_{t}=-1012.718\;a.u.$, whereas for the C_19_Si (B) set $E_{t}=-1012.777\;a.u$. Thus, based on this total energy criteria, the geometrical conformers of the C_19_Si (B) set are more stable than those of the C_19_Si (A) set corroborating the conclusions on the chemical stability made by comparing the values of $E_{HL}$.

Table 1 also brings the relative population, η, of geometrical conformers based on a Boltzmann distribution, at 298.15 K, revealing that the molecules of the C_19_Si (A) group are close to zero while those of C_19_Si (B) are vastly dominant. We expect therefore, that geometrical conformers belonging to the C_19_Si (B) will be more likely produced in a synthesis of C_19_Si. Based on these calculations, it is observed that, in theory, the doping site can be chosen to generate the desired structural and electronic properties for each application. Consequently, the C_19_Si (10) was selected to be studied as a promising molecule to interact with the gas molecules, because C_19_Si (10) showed to be the most stable geometrical conformer belonging to the C_19_Si (B).

### 3.2. Adsorption Energies between C_20_ Fullerene and CO, CO_2_ and N_2_ Molecules

Now let’s turn our attention to the interacting energies between the investigated chemical species. Appendix A presents the respective coordinates of the dimers formatted between the C_20_ and C_19_Si fullerenes and the CO, CO_2_ and N_2_ molecules. In its pristine form, CO, CO_2_ and N_2_ molecules are adsorbed in a parallel orientation as shown in the molecular representations of the dimers in Appendix A. The BSSE-corrected adsorption energy, $E_{ads}$, obtained through Equation (1), portrayed in Figure 3 and in Appendix A.

From Figure 3 we note that both theoretical models point to the C_20_-CO_2_ as the most interacting structure with highest value of $E_{ads}$ among the analyzed dimers. Interaction energies of C_20_-CO and C_20_-N_2_ are dissimilar, so there is no clear selectivity towards these two molecules. Our theoretical results indicate that the complexes studied in the present work are about twice as stable as the complexes investigated by Vessally et al. [33], which can be attributed to both the basis set and the functional chosen for the calculation.

To verify the sensitivity of C_20_ in relation to the adsorption of CO, CO_2_ and N_2_ molecules, the energy gap, $E_{HL}$, of the dimers formed after the adsorption of the gases, was compared with its value prior to the adsorption of the gases. The results of this comparison are presented as a percentual change, represented by $\Delta E_{HL}$. The results of $\Delta E_{HL}$ can be seen in Appendix A. In general, it is observed that the HOMO-LUMO gap is nearly insensitive the presence of the interacting gases with $\Delta E_{HL}\sim 0$ in the absence of any EOEF. Thus, C_20_ fullerene is not sensitive for the detection of CO, CO_2_ and N_2_ molecules.

### 3.3. Electric Field Effect on the Adsorption Energies

To investigate the influence of the electric field, in both structural and energetic characteristics, adsorption calculations for the doped systems were performed. Following we report our findings following the increase of F.
(2)F=0.000 a.u.

The results indicate that the adsorption energies, when compared to the $E_{ads}$ values of the dimers formed with the gases and the pristine fullerenes, increased for the adsorption of carbon monoxide and carbon dioxide. For C_19_Si-N_2_, a decrease in the energy module $E_{ads}$ is observed when the calculation is performed with the level ωB97XD/6-31G(d).

For ease of interpretation, the energy values for the second theoretical level are shown in parentheses. The results show that $E_{ads}$ increases to −0.703 eV (−0.738 eV) for C_19_Si-CO and −0.068 eV (−0.077 eV) for C_19_Si-CO_2_. $E_{ads}$ obtained with ωB97XD/6-311+G(d,p), resulted in a slight increase in the interaction between nitrogen gas and doped fullerene ($E_{ads}$ values observed for the C_19_Si-N_2_ dimer were −0.017 eV (−0.045 eV)). As can be seen in the Figures in Appendix A for C_19_Si-CO, the CO reorients to interact in a perpendicular configuration. For C_19_Si-CO_2_ and C_19_Si-N_2_, the molecules remain interacting in a parallel orientation.

Since the CO molecule has a permanent dipole moment, in which the carbon atom has a positive charge and the oxygen atom has a negative charge, the interaction between the CO molecule and C_19_Si is expected to be the most favorable among the three chemicals analyzed. This is most likely due to the nucleophilic behavior observed on the Si atom in C_19_Si fullerene. This charge accumulation on the heteroatom favors the adsorption of CO in the doped cage through a directional dipolar interaction. The results for the doped system are detailed in Appendix A and in the Figures in Appendix A.

The percentage variation of the energy gap between the frontier orbitals, $\Delta E_{HL}$, indicates that C_19_Si is highly sensitive to the detection of carbon monoxide, and, as can be seen in Appendix A, $\Delta E_{HL}\sim 20\%$ for the two theoretical levels. The negative sign indicates a reduction in the energy gap. For CO_2_ and N_2_, $\Delta E_{HL}$ values show no significant changes, i.e., C_19_Si is not sensitive to the presence of CO_2_ and N_2_.
(3)F≠0.000 a.u.

As can be seen in Appendix A and in Figure 4, for both calculation levels, the $E_{ads}$ values for F=0.001 a.u. are very similar to the $E_{ads}$ values in the absence of the electric field. From the Figures in Appendix A, it is noted that the effects of the electric field, F=0.001 a.u., is to reorient the CO_2_ and N_2_ molecules, so that they interact with a pentagonal face of the C_19_Si cage, which has a more electrophilic character (see Figure 1E). Appendix A shows that as the electric field increases in intensity, the energy gap reduces correspondingly for all dimers, and this effect is more pronounced when C_19_Si absorbs CO_2_ and N_2_. The ωB97XD/6-311+G(d,p) derived $\Delta E_{HL}$ values, with the strongest electric field F=0.025 a.u., of the C_19_Si-CO_2_ and C_19_Si-N_2_ dimers is very close to the variation observed for the C_19_Si-CO dimer. However, even for this strong F, C_19_Si presents greater sensitivity for the detection of the CO molecule.

Now let’s discuss the EOEF effect on the adsorption energies for the doped fullerene with F the range of 0.005 a.u. and 0.025 a.u., presented in Figure 4. We note from Appendix A and Figure 4 that F and $E_{ads}$ show direct proportionality, and for the C_19_Si-CO system, this increase appears almost linear. On the other hand, for the C_19_Si-CO_2_ and C_19_Si-N_2_ systems, F affects $E_{ads}$, with a prominent parabolic behavior as corroborated by fitting correlation polynomial expressions on the $F\;\mathrm{vs}.\;E_{ads}$ portrayed in Figure 4. Clearly, such correlation equations are valid only within the range of F considered in this work, so that even further increasing in the electric field tends to decrease the separation of the molecules and eventually steric repulsions start to dominate.

### 3.4. Intermolecular Interaction Characterization under EOEF Influence

To investigate the character of the intermolecular interactions, we employed QTAIM and RDG analyses. QTAIM analysis allows for investigation of the nature of intra/intermolecular interactions relying on topological properties of the electron density. The use of QTAIM analysis has been successful in characterizing and describing interactions in various chemical systems [60,61,67,68,69,70].

The results obtained for the pure and doped dimers are shown in Appendix A. It is observed that, for all dimers in pure form, with F=0, the values of the electron density, $\rho_{BCP}$, are in the order of $10^{-3}e/a_{0}^{3}$. The Laplacian values of electron density, $\nabla \rho_{BCP}^{2}$, always have positive values. Furthermore, the ratio of kinetic energy density to potential energy density, $\left|G_{BCP}/V_{BCP}\right|$ have values greater than one unit. These results of $\rho_{BCP}$, $\nabla \rho_{BCP}^{2}$ and $\left|G_{BCP}/V_{BCP}\right|$ indicate that C_20_-CO, C_20_-CO_2_ and C_20_-N_2_ dimers are stabilized primarily through van der Waals (vdW) interactions. The RDG scatter plot and the isosurfaces presented in the Figures in Appendix A corroborate these observations.

For doped fullerenes, as can be seen in Appendix A, the dimers C_19_Si-CO_2_ and C_19_Si-N_2_ continue to present values of $\rho_{BCP}$, $\nabla \rho_{BCP}^{2}$ and $\left|G_{BCP}/V_{BCP}\right|$ consistent with observed values for vdW interactions [71,72,73,74,75,76,77,78]. These results remain for the two theoretical levels and for F<0.025 a.u. The RDG scatter plots, for both calculation levels, confirm the non-covalent character of vdW-type observed by the QTAIM analyses. When *F* = 0.025 a.u, the QTAIM parameters and the RDG plot indicate that the interaction changes from vdW to a dipolar character for C_19_Si-CO_2_. When C_19_Si-N_2_ is considered, the interaction between the N_2_ atom and the doped is still of vdW-type.

For the C_19_Si-CO dimer, the results for the two theoretical levels indicate higher values of $\rho_{BCP}$, (about $10^{-2}e/a_{0}^{3}$), for all values electric field. The values of $\nabla \rho_{BCP}^{2}$ remain positive and $\left|G_{BCP}/V_{BCP}\right|<1$. According to the characterization of the QTAIM parameters [71,72,73,74,75,76,77,78], C_19_Si-CO shows a dipolar interaction character, at the ωB97XD/6-31G(d) level and a polar-covalent character for the ωB97XD/6-311+G(d,p) level when F<0.0150 a.u.. Beyond *F* = 0.015 a.u., C_19_Si starts to show a tendency to interact with CO through an interaction with partial polar-covalent character for both theoretical levels. This observation agrees with the characteristics observed in the MEP of C_19_Si (Figure 1), with the $E_{ads}$ values obtained for C_19_Si-CO (Appendix A) and with the directionality observed for the interaction between CO and C_19_Si shown in Appendix A.

An interesting fact was observed in the analysis of QTAIM in the C_19_Si-CO system with the theoretical level ωB97XD/6-31G(d). For this complex, the presence of critical degenerate points was observed in the intermolecular region between the Si and O atoms, as can be seen in Appendix A. This type of degeneracy is common in stability studies, especially in structures under the influence of an external factor such as temperature or in solvents [79,80,81,82]. Degenerate critical points are usually associated with unstable interactions [83]. The closer the RCP and the BCP are, the more unstable the interaction tends to be. In unstable interactions, a small energy disturbance can cause migration from an RCP to a BCP, which leads to the disappearance of RCP [83]. However, when the basis set was extended, this degeneracy disappears.

## 4. Conclusions

With the interest of investigating the adsorption of carbon monoxide, CO, carbon dioxide, CO_2_, and nitrogen gas, N_2_, the C_20_ fullerene was Si-doped. To investigate the impact of heteroatom positioning in the C_20_ cage, twenty C_19_Si were produced. In the present investigation, the Si substitution position influenced the structural and electronic characteristics of the doped system. This led to the formation of two groups of geometrical conformers, namely C_19_Si (A) and C_19_Si (B), classified according to their deformations. After an energetic analysis, it was observed that only the geometrical conformer of the C_19_Si(B) group are energetically favorable at the temperature of 298.15K.

The adsorption of CO, CO_2_ and N_2_ was not favorable to the C_20_ fullerene cage from the energetic point of view. However, it can be observed that the doping with a Si atom enabled the adsorption of the chemical species mentioned to the C_19_Si cage. This adsorption was intensified for all molecules aided by an externally oriented electric field. However, the formation of the C_19_Si-CO dimer was the most favored. Thus, C_19_Si has good selectivity in the adsorption of CO over CO_2_ and N_2_. Based on energetic and topological parameters, when the intensity of the EOEF is smaller than F<0.015 a.u., physisorption take place between CO and C_19_Si. Further increasing F, the adsorption shifts to a polar-covalent character as revealed by QTAIM parameters.

Adsorption energies shows a quadratic relationship with the EOEF with C_19_Si-CO_2_ and C_19_Si-N_2_ dimers, and a linear dependence in the case of C_19_S-CO. Obtaining the correlations for the interaction of these molecules with C_19_Si enable the ration use of EOEF to capture gaseous chemical species. DFT calculations revealed that C_19_Si fullerene is good prototype to selectively detection of carbon monoxide. Modulating the intensity of the EOEF C_19_Si displays is suitable for a CO uptake and detection.

## Figures and Tables

**Figure 1 molecules-27-02968-f001:**
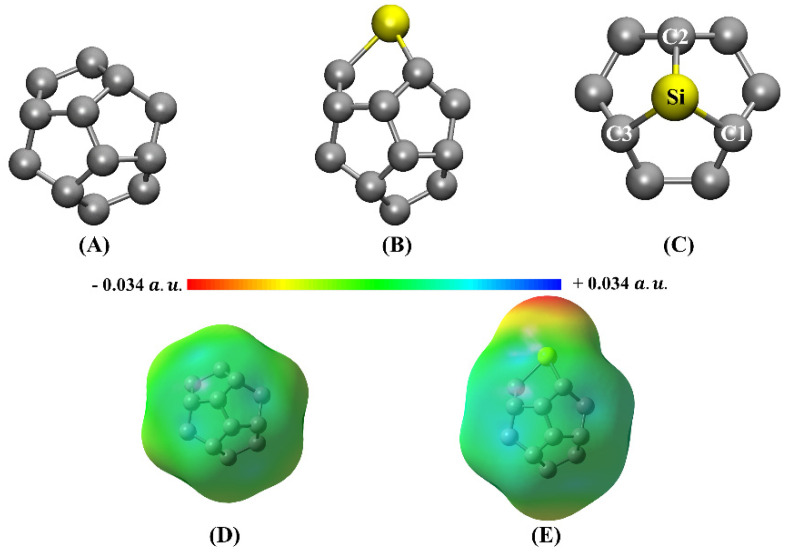
Optimized geometries of the (**A**) C_20_ and (**B**) C_19_Si fullerenes as obtained at the ωB97XD/6-311+G(d,p) level of theory. (**C**) brings an upper view perspective of the C_19_Si structure. Molecular Electrostatic Potential (MEP) map of (**D**) C_20_ and (**E**) C_19_Si.

**Figure 2 molecules-27-02968-f002:**
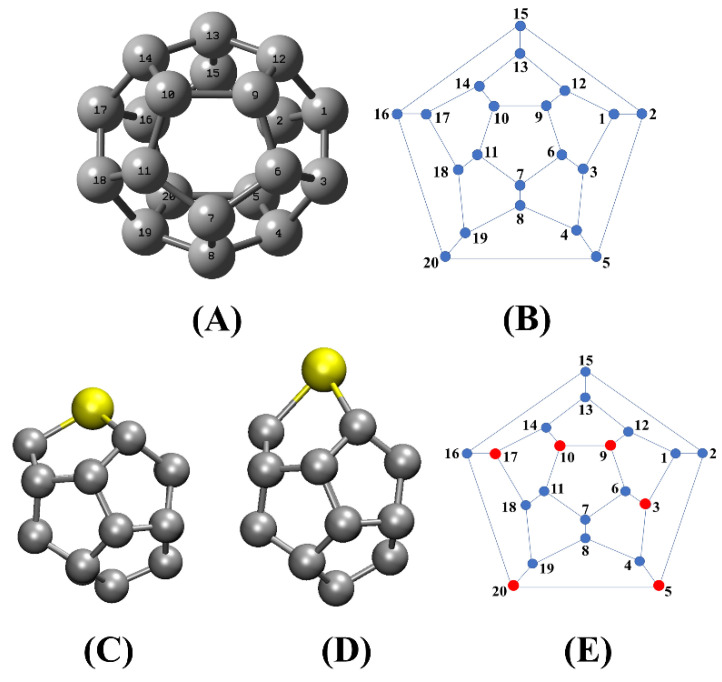
(**A**) C_20_ fullerene structure and (**B**) its corresponding Schlegel diagram. The atom indexes are presented to elucidate the Si doping positions. (**C**) represents the substitutions which produced the set C_19_Si (**A**) formed by: C_19_Si (1), C_19_Si (2), C_19_Si (4), C_19_Si (6), C_19_Si (7), C_19_Si (8), C_19_Si (11), C_19_Si (12), C_19_Si (13), C_19_Si (14), C_19_Si (15), C_19_Si (16), C_19_Si (18) and C_19_Si (19). (**D**) represents the substitutions which produced the set C_19_Si (**B**) formed by: C_19_Si (3), C_19_Si (5), C_19_Si (9), C_19_Si (10), C_19_Si (17) and C_19_Si (20). (**E**) In the Schlegel diagram, the blue dots represent the position of the heteroatoms that resulted in geometrical conformers with the characteristics shown in C_19_Si (**A**). The red dots indicate the positions of the heteroatoms that resulted in geometrical conformers with the characteristics shown in C_19_Si (**B**).

**Figure 3 molecules-27-02968-f003:**
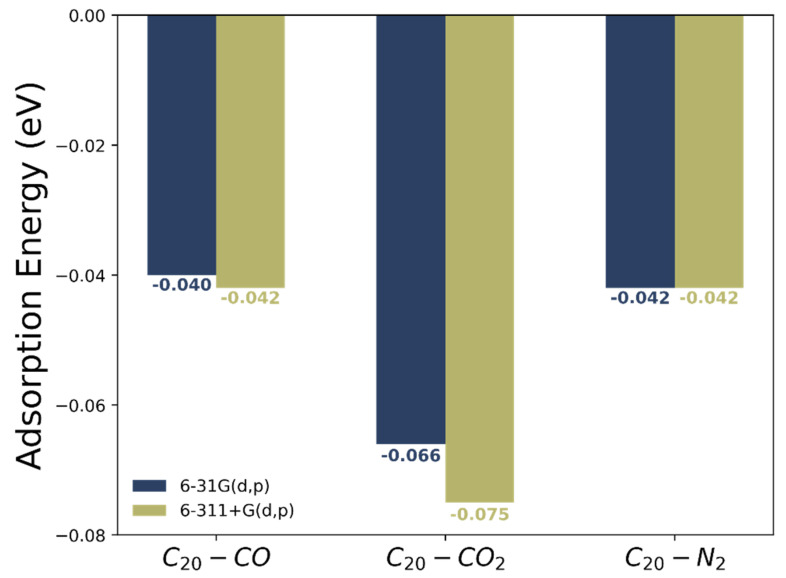
Adsorption energies (in eV), $E_{ads}$, for the dimers formed between C_20_ fullerenes with CO, CO_2_ and N_2_ molecules calculated with the theoretical levels ωB97XD/6-31G(d) and ωB97XD/6-311+G(d,p).

**Figure 4 molecules-27-02968-f004:**
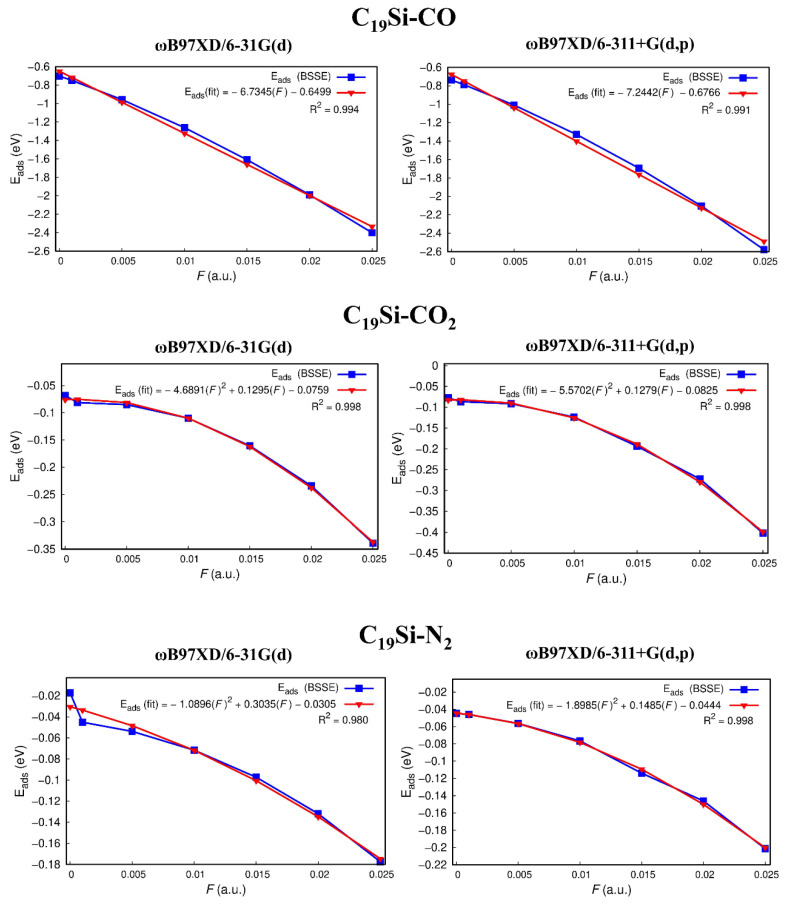
Theoretical, $E_{ads}$, and fitted, $E_{ads}\left(fit\right)$, adsorption energies as a function of the electric field, F.

**Table 1 molecules-27-02968-t001:** MK charges on the Si-atom, total energy, $E_{t}$, (in a.u.) HOMO energy, $E_{H}$, LUMO energy, $E_{L}$, and HOMO-LUMO energy gap $E_{HL}$, (in eV) also shown for discussion in the text. The last column presents the relative population, η in %, of each of the geometrical conformers calculated using the Boltzmann distribution at 298.15 K.

	MK	$E_{t}$	$E_{H}$	$E_{L}$	$E_{HL}$	η
C_19_Si (1)	1.033	−1012.718	−7.237	−2.439	4.798	~0
C_19_Si (2)	1.044	−1012.718	−7.237	−2.439	4.798	~0
C_19_Si (3)	−0.174	−1012.777	−7.765	−2.047	5.718	16.784
C_19_Si (4)	1.033	−1012.718	−7.237	−2.439	4.798	~0
C_19_Si (5)	−0.173	−1012.777	−7.763	−2.047	5.716	16.590
C_19_Si (6)	1.044	−1012.718	−7.237	−2.438	4.799	~0
C_19_Si (7)	1.034	−1012.718	−7.237	−2.436	4.801	~0
C_19_Si (8)	1.034	−1012.718	−7.237	−2.436	4.801	~0
C_19_Si (9)	−0.185	−1012.777	−7.763	−2.048	5.716	16.555
C_19_Si (10)	−0.185	−1012.777	−7.763	−2.047	5.716	16.590
C_19_Si (11)	1.033	−1012.718	−7.237	−2.439	4.798	~0
C_19_Si (12)	1.033	−1012.718	−7.237	−2.447	4.790	~0
C_19_Si (13)	1.045	−1012.718	−7.237	−2.436	4.801	~0
C_19_Si (14)	1.044	−1012.718	−7.237	−2.439	4.798	~0
C_19_Si (15)	1.045	−1012.718	−7.237	−2.436	4.801	~0
C_19_Si (16)	1.033	−1012.718	−7.237	−2.447	4.790	~0
C_19_Si (17)	−0.178	−1012.777	−7.765	−2.047	5.718	16.784
C_19_Si (18)	1.044	−1012.718	−7.237	−2.439	4.798	~0
C_19_Si (19)	1.044	−1012.718	−7.237	−2.439	4.798	~0
C_19_Si (20)	−0.162	−1012.777	−7.765	−2.044	5.722	16.696

## Data Availability

The data presented in this study are available in the article and in the Appendix A.

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
