# Peer review of "Harnessing Greenhouse Gases Absorption by Doped Fullerenes with Externally Oriented Electric Field"

_molecules, 2022, doi:10.3390/molecules27092968_

Round 1
Reviewer 1 Report
In this manuscript, the interaction of fullerene C20 clusters with molecules of carbon monoxide, CO, carbon dioxide, CO2 and nitrogen gas, N2 is theoretically considered. The authors considered in detail the issues of quantum chemical calculations of the studied dimers. A significant number of basis sets were used to verify the calculations.
However, the disadvantage of such studies is that the presented calculations have little practical application. At least there are a lot of water molecules in the air. However, the authors did not mention the water molecule in the manuscript. In this regard, I also believe that the title of the work needs to be adjusted. In my opinion, the manuscript should have the title "Investigation of the influence of an external electric field on the adsorption energy of CO, CO2 and N2 molecules on C20 and C19 Si fullerenes".
Remarks.
L15 The term "alloying" is used unsuccessfully here and further in the text. The term "doping", in my opinion, cannot be used when introducing a single atom into the cluster structure. The term "modification" of the cluster or "decoration" of the cluster is most appropriate here.
L16-17. This is not a new statement. Any change in the structure and composition of a cluster with only 20 atoms will lead to a change in its properties
L22-23. I do not agree with the statement of the covalent nature of the connection. The activation of the adsorption energy by an electric field is investigated. The calculated values of the adsorption energy have an electrostatic nature due to the dipole-dipole interaction.
L75-76. The phrase needs to be corrected. The paper studies the quantum-chemical interaction of a gas molecule with a fullerene cluster. Gas sensors are not studied in the work.
L84-86. The phrase requires a reference to a literary source.
L111 QTAIM -the abbreviation is not deciphered.
L115 It is necessary to decipher the variables in expression (1).
L117-119. It is necessary to clarify in fractions of what the values of F. are given .
L253 BSSE -the abbreviation is not deciphered.
Reviewer 2 Report
The present article is interesting, but needs to be improved.
1. In the Abstract, the first phrase has to be less ambigous: "The present work was dedicated both to the theoretical investigation of the effects caused by the doping of C20 fullerene, with Si silicon atom, and to investigate the adsorption of CO, CO2 and N2 gases to the C20 and C19Si fullerenes".
2. Also, it is not clear what C19Si is: a powder, a solid, a film, pieces, etc.
3. What is "B3LYP/6-31G(d)"? What is "H2@C15M5"?
4. In the Conclusions: "With the interest of investigating the adsorption of carbon monoxide, CO, carbon dioxide, CO2, and nitrogen gas, N2, the C20 fullerene was Si-doped. To investigate the impact of heteroatom positioning in the C20 cage, twenty C19Si were produced.". It is too eliptic: "twenty C19Si were produced". Samples? What kind of material is C19Si? I see the article more theoretic, therefore I do not understand how and where the samples were prodeuced and investigated.
Reviewer 3 Report
The article is dedicated to the study of doped fullerenes C19Si for the adsorption of greenhouse gases such as carbon monoxide (CO) and carbon dioxide (CO2). The subject matter of the manuscript addresses an important problem related to global warming and the effects of climate change, so this research is of interest to many scientists and is sure to find a wide readership.
The manuscript is well-written and keeps editorial standards. The material is presented in a very clear way. The introduction provides scientific background and all necessary information about the material; motivation seems to be fully justified. The experimental methodology is sufficient. The quality of the graphics is well.
I recommend accepting the article after a few minor corrections, as listed below:
- The authors should provide a discussion of the reasons for obtaining two different conformations (C19Si (A) and C19Si(B)), which would explain why the position in which the impurity is inserted into the fullerene cage generates a structural and electronic variation leading to different conformations.
- The sentence in line 315, "We collected graphically in Figure 4." seems to be not complete and should be corrected.
- In line 316, there should be space after "????".
- The abbreviation QTAIM, occurring in line 111, was not explained in the preceding text. Only in Supplementary Materials, the abbreviation was explained. The full name of the theory marked with this abbreviation should be added to the main text.
- In line 384, there is probably an unnecessary dot after the letter F. It should be corrected.
- In figure 4, the fonts at the axes and in the legend are probably too small. I recommend reconsidering enlarging the fonts.
Round 2
Reviewer 1 Report
The authors have made a revision of the manuscript.
However, I cannot agree with the authors' response to Reviewer 1 comments.
It is known that the influence of water molecules has a very strong effect on the adsorption of molecules of any gases on the surface of the adsorbent. Water molecules are strongly polar, so their influence is very great. Water molecules are among the first to be adsorbed on the surface of the adsorbent. Gas molecules need to interact with both the water molecule and the surface of the adsorbent. For an example of the influence of water molecules, I can give, for example, an article [https://doi.org/10.3390/chemosensors6030039 ]. Your manuscript discusses the process of adsorption of gas molecules that make up air, but in the absence of water molecules. I think you should be told in the manuscript that the research is carried out in an atmosphere of dry air.
As for changing the title of the manuscript, I believe that the name I proposed (Investigation of the influence of an external electric field on the adsorption energy of CO, CO2 and N2 molecules on C20 and C19 Si fullerenes) is more correct.
Reviewer 2 Report
Now, the authors have performed the corrections and the article can be published.